# The Role of Bone Marrow Stromal Cell Antigen 2 (BST2) in the Migration of Dendritic Cells to Lymph Nodes

**DOI:** 10.3390/ijms26010149

**Published:** 2024-12-27

**Authors:** Sehoon Park, Eunbi Yi, Jaemyeong Jeon, Jinsoo Oh, Zhengmei Xu, Se-Ho Park

**Affiliations:** College of Life Sciences and Biotechnology, Korea University, 145 Anam-ro, Seongbuk-gu, Seoul 02841, Republic of Korea; shoon153@naver.com (S.P.); pinelime@naver.com (E.Y.); woaudasd@korea.ac.kr (J.J.); oh_jinsoo@korea.ac.kr (J.O.); jungmi69512@naver.com (Z.X.)

**Keywords:** bone marrow stromal antigen 2 (BST2), dendritic cells, adhesion molecules, cell migration, chemokine receptor 7 (CCR7), intercellular adhesion molecule 1 (ICAM-1)

## Abstract

Bone marrow stromal antigen 2 (BST2) is a host-restriction factor that plays multiple roles in the antiviral defense of innate immune responses, including the inhibition of viral particle release from virus-infected cells. BST2 may also be involved in the endothelial adhesion and migration of monocytes, but its importance in the immune system is still unclear. Immune cell adhesion and migration are closely related to the initiation of immune responses. In this study, we found that the expressions of the lymph node homing marker chemokine receptor 7 (CCR7) and an adhesion molecule intercellular adhesion molecule 1 (ICAM-1) in conventional dendritic cells (cDCs) were associated with BST2 expression. Interestingly, *Bst2*^−/−^ cDCs showed lower chemotactic ability, including velocity and accumulative distance toward chemokine ligand 19 (CCL19) gradient in vitro, compared to wild-type cDCs. *Bst2*^−/−^ cDCs also showed reduced migration and reduced retention capacity in draining lymph nodes in vivo. As a result, *Bst2*^−/−^ cDCs as antigen-presenting cells induced lower antigen-specific B cell and T cell responses compared to *Bst2*^+/+^ cDCs. Notably, mice administered the influenza vaccine via *Bst2*^−/−^ cDCs exhibited substantially inefficient virus clearance compared to mice administered the *Bst2*^+/+^ cDCs vaccine. Therefore, we propose that BST2, which plays a critical role in the effective migration and retention of cDCs, is involved in the development of optimal immunological effects in draining lymph nodes.

## 1. Introduction

BST2 (bone marrow stromal cell antigen 2, also known as CD317, tetherin, or HM 1.24 antigen) is a novel type II transmembrane protein consisting of an N-terminal cytoplasmic tail [1] followed by a transmembrane domain, a coiled-coil ectodomain, and a C-terminal glycosyl-phosphatidylinositol (GPI) anchor [2]. In the innate immune response, BST2 is generally expressed in a variety of immune cells such as bone marrow stromal cells, differentiated B cells, T cells, and dendritic cells, and plays a critical role in effective tethering to block virion release by forming dimers or multimers [3,4,5]. Beginning with the studies of protection against HIV-1 by BST2 as tetherin [6], this host factor was thought to prevent the production of other enveloped viruses, such as retroviruses and orthomyxoviruses containing influenza A virus [7,8,9].

In mice, BST2 is highly expressed in plasmacytoid dendritic cells (pDCs) and may be a promising target for delivering antigens to pDCs. Ovalbumin (OVA) immunization in combination with anti-BST2 antibodies and poly(I:C) has been shown to enhance adaptive immunity, thereby improving B16-OVA tumor rejection [10]. Consequently, antigen targeting to pDCs in combination with TLR agonists has been suggested as an effective vaccination strategy [10]. BST2 also promotes dendritic cell (DC) activation during acute Friend retrovirus (FV) infection by improving MHC class II antigen presentation compared to *Bst2^−/−^* DCs. Thus, BST2-expressing DCs from FV-infected mice stimulate FV-specific CD4^+^ T cells more strongly than *Bst2^−/−^* DCs [11]. These data suggest that BST2 may play a key role in enhancing immune responses.

Another proposed role for BST2 is in cell–cell contact, where BST2 on U937 monocyte cells forms intercellular dimers with BST2 on the IFNγ-stimulated endothelium [12]. Lee et al. suggested that IFNγ accelerates wound healing through the BST2-mediated adhesion of endothelial cells [13]. The role of BST2 in cell–cell interaction has also been implicated in tumor development. In breast cancer, the migration and invasion abilities of cancer cells are significantly increased when they exhibit higher BST2 expression [14,15]. It has been suggested that BST2 mediates the adhesion of monocytes to the endothelium through a “tethering” function, similar to the way it inhibits the release of viral particles [12]. The regulation of NF-κB signaling and its downstream proteins by BST2 has also been implicated in the metastasis of tumor cells, as shown by the decreased migration observed when BST2 is downregulated in gastric cancer cells [16]. These studies clearly indicate that BST2 affects the migration of several cell types, although the detailed mechanism remains unclear.

DCs are professional antigen-presenting cells that capture antigens at local entry sites and migrate to draining lymph nodes (LNs) to deliver antigens to adaptive immune cells such as CD4 T cells. In particular, chemokine receptors like CCR7 and adhesion molecules such as ICAM-1 and VCAM-1 are crucial for the migration of antigen-captured DCs to draining LNs. Chemokines act as mediators of cell migration during both inflammation and immune surveillance under steady-state conditions [17]. The expression of chemokine receptors like CCR7 is upregulated by the TLR4-mediated stimulation of DCs [18], and the administration of TLR4-agonist LPS promotes the in vivo mobilization of DCs from peripheral tissues to draining LNs within a few hours [19,20]. Therefore, CCR7 is critically important for DC migration and T cell-dependent immunity [21].

Collagen-induced arthritis (CIA) is an experimental animal model for human rheumatoid arthritis (RA). Autoreactive T cells and antibodies are present in both human RA and CIA. The high incidence of anti-type II collagen (CII) antibodies and CII-specific T cells is common in human RA and CIA, indicating that type II collagen is a prominent autoantigen [22]. These autoreactive responses are induced by the activation of CD4 T cells and B cells responding to autoantigens [23]. In this context, the ability of conventional DCs (cDCs) as professional antigen-presenting cells is critical for the effective induction of arthritis.

In this study, we found that *Bst2^−/−^* cDCs expressed lower levels of CCR7, ICAM-1, and LFA-1 compared to *Bst2^+/+^* DCs. *Bst2^−/−^* DCs also showed reduced chemotactic migration toward the CCR7 ligand CCL19 in vitro, along with decreased migration in vivo. As a result, *Bst2^−/−^* cDCs exhibited relatively insufficient adaptive immune responses, including B and T cell immunity, and a reduced ability to eliminate influenza virus infection. Therefore, we suggest that BST2 plays an important role in the development of adaptive immune responses, particularly during the initiation stage, by enhancing DC migration and retention in draining LNs.

## 2. Results

### 2.1. BST2 Is Associated with the Expression of the Chemokine Receptor CCR7 and Adhesion Molecules ICAM-1 on cDCs

BST2 expression is normally maintained at low levels in most immune cells, including B cells, T cells, and DCs, but can be upregulated upon cell activation by molecules like lipopolysaccharide (LPS) (Figure 1A). However, DCs from BST2^−/−^ mice exhibited no sign of BST2 upregulation (Figure 1B). *Bst2^−/−^* mice did not show any notable differences in the cellularity of immune cells, including total cell numbers and sub-populations in the bone marrow (BM), thymus, spleen, and liver compared to *Bst2^+/+^* mice Appendix A.

Dendritic cells are important for initiating the adaptive immune response through antigen capture and migration to draining LNs. To investigate the migration-related phenotype of DCs, splenocytes from *Bst2^+/+^* and *Bst2^−/−^* mice were stimulated with LPS, and surface markers related to cell adhesion and migration were analyzed. Although there were no significant differences in most markers analyzed (Appendix A), chemokine receptor CCR7 and adhesion molecule ICAM-1 expression were substantially lower in *Bst2^−/−^* cDCs. Other cell types, such as B and T cells, however, did not show BST2 dependency for CCR7 and ICAM-1 expression. pDCs, which normally express the highest levels of BST2, also did not show any BST dependency for CCR7 and ICAM-1 expression. LFA expression in all cell types was not affected by the presence or absence of BST2 expression (Figure 1C–E). These results suggest that BST2 is involved in the expression of CCR7 and ICAM-1, particularly in cDCs.

### 2.2. BST2 Expression on cDCs Promotes CCR7-Mediated Chemotaxis

To investigate whether BST2 on cDCs is involved in CCR7-mediated migration, we conducted a chemotaxis assay using CCL19, the ligand for CCR7. To distinguish Bst2^+/+^ and Bst2^−/−^ cDCs, purified cDCs were labeled with either CFSE or CMTMR, mixed in a 1:1 ratio, and allowed to migrate toward CCL19 in the opposite chamber. When we analyzed the trajectory of cDCs, we observed that the overall movement of Bst2^−/−^ cDCs was reduced compared to that of Bst2^+/+^ cDCs (Figure 2A). To confirm that cDCs migrated toward the CCL19 gradient, we measured the forward migration index in the y-direction (y forward migration index, yFMI) (Figure 2B) and found that Bst2^+/+^ cDCs migrated more strongly to CCL19 than Bst2^−/−^ cDCs.

To analyze cell mobility in more detail, we measured the center of mass displacement at two-minute intervals, which is the straight-line distance traveled by cells. We found that the straight-line distance of Bst2^−/−^ cDCs was shorter than that of Bst2^+/+^ cDCs (Figure 2C). We also compared accumulated distances (Figure 2D) and found a slight but noticeable difference, with Bst2^+/+^ cDCs showing greater cumulative movement. These results indicate that Bst2^−/−^ cDCs were less responsive to CCL19 and moved more randomly, likely due to their relatively lower CCR7 expression compared to Bst2^+/+^ cDCs. Together, these findings suggest that BST2 may play a role in cDC migration by facilitating CCR7-mediated chemotaxis.

### 2.3. BST2 Expression on cDCs Facilitates Their Migration into Draining Lymph Nodes

Because the expression of migration-related molecules and CCR7-dependent migration ability were downregulated in *Bst2^−/−^* cDCs, we hypothesized that the migration of cDCs from local sites to draining LNs might be impaired in *Bst2^−/−^* mice. To test this hypothesis, we mixed equal numbers of LPS-stimulated *Bst2^+/+^* and *Bst2^−/−^* cDCs, injected them into the footpads of recipient mice, and monitored the migration of cDCs into the draining popliteal LNs (pLNs). Prior to injection, cDCs were stained with CFSE to distinguish them from recipient DCs (Figure 3A). To exclude any interference from unequal apoptotic events, we analyzed cell viability before transfer and found no significant differences in apoptosis between the groups (Figure 3A).

At 2.5 days post-injection, the population ratio of *Bst2^−/−^* cDCs in the draining LNs was smaller than that of *Bst2^+/+^* cDCs in both *Bst2^+/+^* and *Bst2^−/−^* recipients (Figure 3B,C). In *Bst2^+/+^* recipients, there was a significant increase in the migration of *Bst2^+/+^* cDCs at 2.5 days post-injection compared to 1.5 days post-injection. However, the migration of *Bst2^−/−^* cDCs did not increase over the same time period. Interestingly, in *Bst2^−/−^* recipients, *Bst2^+/+^* cDCs did not show the expected increase in accumulation at 2.5 days post-injection compared to 1.5 days, and *Bst2^−/−^* cDCs showed a drastic decrease in accumulation over the same period (Figure 3B,C). These results suggest that BST2 expression is necessary not only for the migration of cDCs but also for their retention within the LN tissue. Furthermore, these findings imply that BST2 expression in the local LNs is necessary for the prolonged retention of migrated cDCs. We propose that BST2 promotes cell adhesion via intercellular interaction between BST2 ectodomains [12].

### 2.4. BST2-Dependent Migration of cDCs Is Associated with Optimal Immune Responses

Since BST2 expression on cDCs is crucial for efficient migration and retention both in vitro and in vivo, we hypothesized that the loss of BST2 expression on antigen-presenting cDCs could impair antigen-specific immune responses in the draining LNs. To access the antigen-specific T cell response, we transferred OVA323-339-pulsed *Bst2^+/+^* or *Bst2^−/−^* cDCs into each footpad of ovalbumin-specific CD4^+^ T cell transgenic OT-2 mice. T cells in dLN were analyzed 4 days after cDCs transfer. There were no significant differences in the cellularity of dLN following the transfer of either *Bst2^+/+^* or *Bst2^−/−^* cDCs, although both dLNs showed a massive expansion of cells, compared to cDC-untransferred controls (Figure 4A).

Next, cells from the dLNs of mice that received either *Bst2^+/+^* and *Bst2^−/−^* cDCs were re-stimulated in vitro with OVA323-339-pulsed *Bst2^+/+^* cDCs, and antigen-specific T cell proliferation and IFNγ production were compared. T cells primed by *Bst2^−/−^* cDCs showed reduced proliferation and produced less IFNγ compared to those primed by *Bst2^+/+^* cDCs (Figure 4B). These results suggest that the decreased migration and retention of *Bst2^−/−^* cDCs limits optimal TCR recognition. However, if the loss of BST2 expression were to affect the antigen presentation pathway of cDCs, we would expect similar results. To address this, we reactivated *Bst2^+/+^* cDC-primed T cells with either OVA323-339-pulsed *Bst2^+/+^* cDCs or *Bst2^−/−^* cDCs. In this case, both stimulations resulted in similar levels of T cell proliferation and IFNγ production (Figure 4C). This result indicates that the loss of BST2 expression in *Bst2^−/−^* cDCs does not affect antigen presentation.

We then assessed the role of BST2 expression in humoral immunity. *Bst2^+/+^* and *Bst2^−/−^* mice were immunized with the OVA antigen, and OVA-specific IgG and IgM production were measured in the sera one week after boost immunization. We found substantially lower levels of OVA-specific IgG and IgM antibodies in the sera of *Bst2^−/−^* mice compared to *Bst2^+/+^* mice (Figure 4D). Insufficient antibody production in *Bst2^−/−^* mice compared to *Bst2^+/+^* mice may be due to a deficiency in T cell stimulation by *Bst2^−/−^* dDCs, as well as a defect in the *Bst2^−/−^* B cells themselves. Although we did not further analyze the role of *Bst2* in B cells, additional experiments showed that splenocytes from OVA-immunized *Bst2^−/−^* mice produced less IFNγ and IL-4 upon OVA stimulation compared to splenocytes from *Bst2^+/+^* mice (Figure 4E), which suggests at least some involvement of T cells.

These results suggest that BST2 expression on cDCs play a critical role in the migration and retention of cDCs in dLNs, thereby contributing to optimal immune responses including T cell priming and humoral immunity.

### 2.5. BST2-Expressing DCs Strengthen Immune Responses Against Influenza Virus

DCs are professional antigen-presenting cells that play key roles in antiviral and antitumor immunity [24]; when infected with a virus, DCs prevent the replication and release of infectious progeny viruses [25,26]. To assess the role of BST2 in antiviral immunity mediated by DCs, we vaccinated Bst2^+/+^ mice with either Bst2^+/+^ or Bst2^−/−^ DCs that had been infected with mouse-adapted influenza A virus (MA-IAV). The vaccinated mice were then infected with live virus one week after the boost vaccination (Figure 5A). Although weight loss was lower in both DC-vaccinated groups compared to the unvaccinated group, disease course was more severe in the Bst2^−/−^ DC-vaccinated group than in the Bst2^+/+^ DC-vaccinated group (Figure 5B). Next, we measured the levels of anti-MA-IAV antibodies in the sera of the vaccinated mice (Figure 5C). Although slightly more MA-IAV-specific total IgG was produced in Bst2^+/+^ DC-vaccinated group than in the Bst2^−/−^ DC-vaccinated group, there was significantly more antigen-specific IgG2a, a hallmark of Th1 responses, in the Bst2^+/+^ DC-vaccinated group, Conversely, the level of IgG1, a Th2-type isotype, was higher in the Bst2^−/−^ DC-vaccinated group. It is well established that DC-mediated immunization against influenza induces Th1 responses, such as the production of IgG2a, which enables host cells to efficiently eliminate the virus [26]. Additionally, a neutralization assay using sera from the Bst2^+/+^ and Bst2^−/−^ DC-vaccinated mice showed significant differences in the ability to block viral infection (84.7% neutralization by the Bst2^+/+^ DC-vaccinated group vs. 50.5% by the Bst2^−/−^ DC-vaccinated group) (Figure 5D). Lung virus titration, based on a plaque-forming assay, also showed that significantly more virus remained at the infection site in the Bst2^−/−^ DC-vaccinated group than in the Bst2^+/+^ DC-vaccinated group (Figure 5E). These results suggest that the Bst2^+/+^ DC-vaccinated group had better viral clearance than the Bst2^−/−^ DC-vaccinated group. Furthermore, lung histology, using hematoxylin and eosin (H&E) staining, revealed more severe infiltration of nucleated cells and increased interstitial thickness in the lungs of Bst2^−/−^ DC-vaccinated mice compared to Bst2^+/+^ DC-vaccinated mice (Figure 5F). These findings support the conclusion that BST2 expression in cDCs is important for promoting a sufficient humoral immune response, which is essential for protective immunity against viral infections.

### 2.6. BST2-Mediated DC Migration Is Associated with the Development of Effector T Cell Populations

To investigate the development of effector T cells, splenocytes from virus-infected mice were re-stimulated with irradiated MA-IAV-infected *Bst2^+/+^* splenocytes. After 3.5 days of re-stimulation, splenocytes were analyzed by FACS analysis. A smaller population of both CD4^+^ and CD8^+^ effector T cells was observed in the *Bst2^−/−^* DC-vaccinated group compared to the *Bst2^+/+^* DC-vaccinated group, particularly in the CD8^+^ T cell population (Figure 6A).

Since CD8^+^ T cell-secreted IFNγ is critical in efficient effector function after DC vaccination [27], we measured the intracellular cytokine levels of CD8^+^ T cells of isolated from splenocytes of both *Bst2^+/+^* DC- and *Bst2^−/−^* DC-vaccinated mice. Significantly higher levels of intracellular IFNγ were detected in CD8^+^ T cells from the *Bst2^+/+^* DC-vaccinated group compared to the *Bst2^−/−^* DC-vaccinated group (Figure 6B). Additionally, an IFNγ-based enzyme-linked immunospot (ELISPOT) assay revealed more IFNγ-expressing cells in the *Bst2^+/+^* DC-vaccinated group than in the *Bst2^−/−^* DC-vaccinated group upon stimulation with influenza-infected antigen-presenting cells (Figure 6C). The amounts of secreted cytokines in the culture supernatants were measured by enzyme-linked immunosorbent assay (ELISA), showing significantly more IFNγ in the culture media from the *Bst2^+/+^* DC-vaccinated group than from the *Bst2^−/−^* DC-vaccinated group. In contrast, the level of IL-4 was higher in the *Bst2^−/−^* DC-vaccinated group (Figure 6D). Next, we assessed the cytotoxic activity of stimulated splenocytes against influenza-infected EL4 target cells and observed stronger cytotoxicity from the *Bst2^+/+^* DC-vaccinated group compared to the *Bst2^−/−^* DC-vaccinated group (Figure 6E). These results suggest that BST2, which promotes efficient DC migration, enhances the effector function of activated cells, including CD8^+^ T cells, improving their cell-mediated response against the virus.

### 2.7. BST2 Affects the Severity of Autoimmune Diseases in Mice

To investigate the role of Bst2 in the development of autoimmune diseases, we induced arthritis in mice by injecting chicken collagen. Collagen-induced arthritis (CIA) was induced in DBA1/J Bst2^+/−^ and DBA1/J Bst2^−/−^ littermate mice. Our results showed that Bst2^+/−^ mice developed arthritis earlier and with greater severity than Bst2^−/−^ mice, suggesting a role for Bst2 in promoting the development of autoimmune diseases. Additionally, Bst2^+/−^ mice exhibited higher serum antibody levels compared to Bst2^−/−^ mice. These finding suggest that Bst2 may regulate autoantigen presentation by cDCs in the lymph nodes.

## 3. Discussion

BST2 is a type II transmembrane protein with an N-terminal cytoplasmic region and a C-terminal GPI-anchor. It is known to inhibit the release of viral particles from infected cells by tethering the virus to the host cell membrane through dimerization on both sides of the membrane [3,4,5,7]. BST2 also enhances the adhesion of epithelial cells and monocytes [12] and has been implicated in the metastasis and poor prognosis of tumors [14,15,28,29]. Cell adhesion is a critical factor in cell migration [30], which prompted us to investigate whether BST2-mediated adhesion plays a role in the migration of antigen-presenting cells (APCs) from the site of infection to the draining lymph nodes (dLNs). To explore whether BST2 expression influences immune cell development and differentiation, we compared cell distributions in various lymphoid organs of Bst2^+/+^ and Bst2^−/−^ mice. We observed no significant differences in cell populations under naïve or LPS-induced inflammatory conditions (Appendix A). Activation markers, such as CD69 and CD44 on lymphocytes, and B7 family members on macrophages, were also similarly expressed in both groups. However, we found that BST2-deficient mice exhibited substantial downregulation of CCR7 and ICAM-1, particularly on cDCs. These molecules are essential for leukocyte lymph node homing and cell–cell adhesion. Therefore, we hypothesized that BST2 may be involved in immune responses in the draining LNs.

Prior to analyzing the in vivo immune response, we tested whether the reduced expression of CCR7 on Bst2^−/−^ cDCs impaired their migration toward CCL19 in vitro. Bst2^−/−^ cDCs demonstrated reduced chemotaxis in response to CCL19, indicating that BST2 is important for the migration of cDCs to the lymph nodes. Additionally, we observed that Bst2^−/−^ cDCs had significantly lower retention in the lymph nodes compared to wild-type cDCs (Figure 3). These findings suggest that BST2 contributes to both the migration and retention of cDCs in LNs.

To further investigate this, we used various in vivo immunization protocols, including the administration of antigen-loaded cDCs (OVA-pulsed or virus-infected) and the direct injection of protein antigen into peripheral tissues. In all cases, when Bst2^−/−^ cDCs were used or Bst2^−/−^ recipients were recruited, immune responses were substantially impaired (Figure 4, Figure 5 and Figure 6). Therse results further support the hypothesis that BST2 plays a key role in immune cell migration and the activation of adaptive immune responses.

The cytoplasmic tail of murine BST2 consists of 30 amino acids and alternative translation can generate long or short isoforms. The long isoform contains a dual-tyrosine (YxY) motif, which has been proposed as a positive regulator of NF-κB signaling, a pathway involved in MyD88 and TAK-1-mediated signal transduction [31,32,33,34]. NF-κB can regulate the expression of CCR7 and ICAM-1 on bone marrow-derived dendritic cells (BMDCs) and endothelial cells [35,36]. This suggests that BST2-mediated NF-κB signaling could be associated with the regulation of CCR7 expression in cDCs. To test this, we analyzed CCR7 expression on cDCs lacking the N-terminal YxY motif of BST2 (Bst2 ^S/−^). These cDCs, derived from backcrossed NZW mice with a silent mutation in the first methionine codon of the BST2 gene [37], showed significantly reduced CCR7 expression and NF-κB signaling upon LPS stimulation Appendix A. These results suggest that BST2-mediated NF-κB signaling, via the YxY motif, regulates the expression of chemokine receptors and adhesion molecules, which are crucial for cDC migration and LN homing, ultimately facilitating adaptive immune responses.

An insufficient number of APCs and reduced retention in the LNs may limit optimal adaptive immune responses. In summary, our findings suggest that BST2 is critical for the migration of cDCs to the LNs and the priming of naïve lymphocytes, processes essential for effective immune responses to pathogens.

Although BST2^−/−^ DCs exhibited decreased levels of CCR7 and ICAM-1, their ability to activate T cells in vitro was comparable to wild-type DCs (Figure 4C). This suggests that BST2 may not directly influence antigen presentation, but rather play a role in the migration of DCs to LNs, thereby enhancing the priming of naïve T cells.

To further investigate the role BST2 in infectious diseases, we conducted an influenza vaccination experiment using dendritic cells to assess its impact on antigen presentation. Mice receiving influenza antigens via BST2^+/+^ DCs exhibited higher levels of anti-influenza-specific immunoglobulins in serum and less severe disease after virus challenge compared to mice receiving BST2^−/−^ DCs loaded with influenza antigen (Figure 5).

Conversely, we hypothesized that BST2 may influence the development of autoimmune diseases, given that these conditions also rely on autoantigen-specific T cell activation in the LNs. In a CIA model, BST2^−/−^ mice exhibited significantly lower clinical disease scores and reduced serum levels of collagen-specific immunoglobulin compared to BST2^+/−^ mice (Figure 7).

Overall, our study demonstrates that BST2 enhances the delivery of antigens by DCs to the LNs, thereby promoting adaptive immune responses. BST2-deficient DCs exhibit impaired migration and are less efficient in their role as APCs. These findings suggest that BST2 could be a promising therapeutic target for both infectious and autoimmune diseases, with potential strategies including either enhanced expression or selective inhibition. Further research is needed to develop a therapeutic protocol targeting BST2 in disease models.

## 4. Materials and Methods

### 4.1. Cell Lines, Virus, and Mice

The MDCK.2 cell line was cultured in Minimal Essential Media (MEM, Gibco, Grand Island, NY, USA) supplemented with 10% FBS, 2 mM L-glutamine, and 1% penicillin–streptomycin [7]. The EL4 cell line was cultured in Dulbecco’s Modified Eagle’s Medium (DMEM, Gibco) supplemented with 10% FBS, 2 mM L-glutamine, 10 mM HEPES (Sigma-Aldrich, St. Louis, MO, USA), 1% penicillin–streptomycin, 10 μg/mL gentamycin (Gibco), and 50 μM β-mercaptoethanol (Gibco-BRL). Seasonal influenza virus strain A/Brisbane/59/2007 (H1N1) was obtained from the Korea Centers for Disease Control and Prevention (KCDC, Chungcheongbuk-do, Cheongju, Republic of Korea).

The C57BL/6 wild-type (Bst2^+/+^) mice were purchased from Orient Bio (Seoul, Korea). Bst2-knockout (Bst2^−/−^) mice were created on a C57BL/6Tac background by Xenogen Biosciences (Cranbury, NJ, USA) [38], DBA/1 wild-type (Bst2^+/+^) mice were purchased from Charles River Research Laboratories (Charles River Laboratories, Tokyo, Hino, Japan). DBA/1 Bst2^−/−^ mice were generated by backcrossing C57BL/6 Bst2^−/−^ mice with DBA/1 mice for more than eight generations. Ova-specific CD4+ T cell transgenic OT-II mice were obtained from Jackson laboratory. All mice were bred under specific pathogen-free conditions at Korea University. All animal experiments were approved by the Institutional Animal Care and Use Committee of Korea University (KUIACUC-2019-0015).

### 4.2. Mouse Adaptation of H1N1 Influenza Virus

Mouse-adapted influenza virus was obtained from eight consecutive lung-to-lung passages in mice [39]. Briefly, anesthetized individual C57BL/6 mice were given 1 × 10^8^ pfu of wild-type H1N1 (A/Brisbane/59/2007) influenza virus by a nasal route, and lungs were collected two days after virus infection. The lungs were then homogenized, and the supernatant of centrifuged lung lysates was used for the following passages. After a total of 8 passages, the virus from lung lysates was additionally replicated once in MDCK.2 cells.

To prepare a virus stock, MDCK.2 cells plated 20–24 h before infection were inoculated with lung lysates for 30 min and serum-free media containing TPCK-treated trypsin was sequentially added and incubated in the 37 °C, 5% CO_2_ incubator for an additional 48 h. Culture media were harvested and stocked for infection experiments.

### 4.3. Cell Preparation and DC Enrichment

The spleen was gently minced in the nylon mesh and suspended in PBS. The erythrocytes were lysed in ammonium-chloride-potassium (ACK) lysis buffer for 10 min. The splenic DCs were obtained from splenocytes of C57BL/6 *Bst2^+/+^* and *Bst2^−/−^* mice via magnetic-activated cell sorting (MACS) using anti-CD11c microbeads (Miltenyl Biotec, North Rhine-Westphalia, Bergisch Gladbach, Germany). The purity of sorted DCs was more than 90%.

### 4.4. Chemotaxis Assay

Splenic DCs were isolated from C57BL/6 *Bst2^+/+^* mice and *Bst2^−/−^* mice, and were labeled with 0.625 μM of carboxyfluorescein diacetate succinimidyl ester (CFSE) and 10 μM of 5-(and-6)-(((4-chloromethyl)benzoyl)amino)tetramethyl-rhodamine (CMTMR), respectively. DCs were mixed in 1:1 ratio and activated with 1 μg/mL of LPS for an hour then used for the chemotaxis assay with a μ-Slide Chemotaxis (IBIDI USA, Inc., Fitchburg, WI, USA), according to the manufacturer’s protocol. Briefly, cDCs were seeded into the channel of a μ-slide and 1.25 μg/mL of CCL19 was added in the left chamber as a chemoattractant. Images were acquired for an hour with a time-lapse of 2 min intervals in between. The tracking of DCs was analyzed by an ImageJ program (NIH, Bethesda, MD, USA). The Forward Migration Index (FMI), commonly referred to as the chemotactic index, is a measure for directed, chemotactic cell migration. It represents the efficiency of the forward migration of cells. Straight-line distance measures the distance between the start and end point regardless of cells’ travel path. The center of mass represents the average of all single cell endpoints. Its x and y values indicate the direction in which the group of cells primarily traveled.

### 4.5. In Vivo DC Migration Assay

Isolated *Bst2^+/+^* DCs (from CD45.1 congenic mice) and *Bst2^−/−^* DCs (from CD45.2 mice) were labeled with 0.625 μM of CFSE at a mixed 1:1 ratio. Cells were activated with 1 μg/mL of LPS for an hour and washed twice by centrifugation. Mice were anesthetized using 2.5% of Avertin solution and 0.1~1 × 10^6^ cells in 50 μL were injected into a footpad by using a 30 G needle. About 1.5 and 2.5 days later, mice were sacrificed, and popliteal lymph nodes (pLNs) were harvested. Single cells from dissociated lymph nodes were analyzed by FACS staining.

### 4.6. OVA Immunization

A total of 100 μg of OVA proteins were emulsified with CFA (Gibco, Grand Island, NY, USA) and then administered by subcutaneous (s.c.) injection into C57BL/6 *Bst2^+/+^* or *Bst2^−/−^* mice. Two weeks post-injection, we gave them 100 μg of OVA proteins, emulsified with IFA (Gibco, Grand Island, NY, USA), by s.c. injection. Sera or cells from immunized mice were analyzed a week after boosting immunization.

### 4.7. DC Vaccination and MA-IAV Virus Infection

Isolated DCs were incubated with 5 Multiplicity of Infection (MOI) of mouse adapted-influenza A virus (MA-IAV) and 1 μg/mL of LPS from Salmonella typhimurium (Sigma-Aldrich, St. Louis, MO, USA) in a 37 °C, 5% CO_2_ incubator for 45 min. Infected DCs were washed twice by centrifugation and resuspended in PBS. A total of 1 × 10^6^ cells of *Bst2^+/+^* or *Bst2^−/−^* DCs were injected intraperitoneally (i.p.) into C57BL/6 *Bst2^+/+^* recipients and boosted them i.p. two weeks later with an equal number of the same type of DCs. After a week of DC immunization, recipients were anesthetized using 2.5% Avertin solution, and 5 × 10^5^ pfu of MA-IAV were inoculated by the nasal route. The body weights of inoculated mice were measured daily. The mice were sacrificed four days after virus infection.

### 4.8. Splenocyte Re-Stimulation

C57BL/6 *Bst2^+/+^* splenocytes were irradiated at a dose of 1.5 Gy of γ-irradiation and incubated with 0.5 MOI of MA-IAV in a 37 °C, 5% CO_2_ incubator for 45 min, and were used as antigen-presenting cells. The splenocytes isolated from vaccinated mice were cultured with an equal number of irradiated splenocytes in RPMI 1640 medium supplemented with 10% FBS, 2 mM L-glutamine, 1% penicillin–streptomycin, 10 μg/mL gentamycin, and 50 μM β-mercaptoethanol (Gibco-BRL, Grand Island, NY, USA) for 3–4 days.

### 4.9. Plaque Assay

For the titration of lung virus, mice were sacrificed, and four lobes (superior, middle, inferior, and post-caval lobe) of the lung were collected, and the lungs were frozen immediately using liquid nitrogen. The lungs were homogenized for a minute with 1 mL of serum-free MEM containing 0.3% BSA and 1 μg/mL of TPCK-trypsin. Lung lysates were centrifuged at 300× *g* for 10 min at room temperature, and supernatants were transferred into new tubes. A total of 5 × 10^5^ MDCK.2 cells were cultured for 16–18 h in 12-well plates. Monolayer cells were treated with serially diluted lung lysates for 45 min, and then the inoculum was replaced with overlay medium containing 0.8% Seaplaque agarose (Lonza, Basel-Stadt, Basel, Switzerland). After 48 h, the overlay medium was removed, plates were stained with 0.2% crystal violet in 20% ethanol, and plaques were counted.

### 4.10. Neutralization Assay

For the titration of anti-influenza neutralizing antibodies, sera from immunized mice were inactivated at 56 °C for 30 min, then were serially diluted up to 10^−3^. Heat-inactivated sera and virus were incubated for an hour at 37 °C. A total of 5 × 10^5^ MDCK.2 cells were cultured for 16–18 h in 12-well plates. Monolayer cells were treated with incubated viruses for 45 min, and then the inoculum was replaced with overlay medium containing 0.8% Seaplaque agarose (Lonza, Basel-Stadt, Basel, Switzerland). After 48 h, the overlay medium was removed, plates were stained with 0.2% crystal violet in 20% ethanol, and plaques were counted. The percentage of neutralization was calculated with the equation [(positive sample)/positive × 100]. Positive control was obtained by inoculating the virus only.

### 4.11. H&E Staining of Lung Section

Hematoxylin and eosin (H&E) staining of the lung tissue was performed for histopathological analysis. Lung tissues were fixed with 4% paraformaldehyde (PFA) solution and were embedded in paraffin. Paraffin-embedded tissues were sliced into 4 μm sections and mounted on silane-coated slides. Lung sections were dewaxed with xylene and rehydrated in water. Stained slides were examined microscopically for endothelial cell lesions, inflammation-based consolidation, and cell infiltration.

### 4.12. Flow Cytometry (FACS) Analysis

Cells were pre-incubated for 30 min at 4 °C with anti-FcγRII/III antibody (anti-CD16/32, 2.4G2) to prevent non-specific binding of the antibody, then labeled with fluorochrome-conjugated antibodies for 30 min. Cells were stained and washed in FACS buffer (PBS containing 0.1% BSA and 0.01% sodium azide). The following monoclonal antibodies were used: anti-TCRβ-FITC or APC-Cy7 (H57-597), anti-CD3ε-FITC or PE (145-2C11), anti-CD19-FITC (1D3), anti-B220-FITC or PerCP-Cy5.5 (RA3-6B2), anti-NK1.1-FITC (PK136), anti-CD11c-FITC, PE, PerCP-Cy5.5 or APC (HL3), anti-H-2Kb-FITC (AF6-88.5), anti-CD86-FITC (GL1), anti-CD45.1-PE (A20), anti-CD8α-PE or PerCP-Cy5.5 (53-6.7), anti-I-Ab-PE (AF6-120.1), anti-IFNγ-PE (XMG1.2), anti-CD44-PerCP-Cy5.5 or biotin (IM7), anti-CD62L-APC (MEL-14), anti-IL-4-APC (11B11), anti-CCR7-PE-Cy7 (4B12), anti-LFA-1-PE-Cy7 (2D7), anti-CD11b-APC-Cy7 (M1/70), anti-ICAM-1-biotin (3E2), anti-CD69-biotin (H1.2F3), anti-CD80-biotin (16-10A1), anti-PDCA-1-biotin, (e.Bio927), and anti-Vα2-PE (B20.1), anti-Vβ5.1-FITC (MR9-4), streptavidin-APC or PE-Cy7 (BD Biosciences, San Jose, CA, USA). Stained cells were analyzed on FACSCalibur and FACSVerse flow cytometers (BD Biosciences) using FlowJo v10 software (FlowJo, LLC, Ashland, OR, USA).

### 4.13. Intracellular Cytokine Analysis

For the detection of intracellular cytokines, re-stimulated splenocytes were treated for the final 6 h of incubation with GolgiStop (BD Pharmingen, San Diego, CA, USA). The surface markers of cells were first stained with appropriated monoclonal antibodies, then fixed, and permeabilized using a Cytofix/Cytoperm kit (BD Pharmingen, California, San Diego). They were finally stained with APC-conjugated anti-IFNγ and anti-IL-4 monoclonal antibodies for 45 min on ice. Intracellular cytokine-stained cells were analyzed on a FACSVerse flow cytometer (BD Biosciences, San Jose, CA, USA) using FlowJo software (FlowJo, LLC, Oregon, Ashland)

### 4.14. In Vitro Cytotoxicity Assay

As target cells, a C57BL/6 thymoma cell line, EL4 cells, were incubated with 1 MOI of MA-IAV in a 37 °C, 5% CO2 incubator for 2 h, and were washed by centrifugation. For the calcein-AM assay, target cells were labeled with 30 μM of calcein-AM (Thermo Fisher Scientific, Waltham, MA, USA) solution at 37 °C for 1 h, washed two times, distributed to 96-well round-bottom plates (5 × 10^3^ cells/well), then incubated with specific ratios of re-stimulated effector cells for 3 h. Supernatants were harvested by centrifugation, and calcein-AM release was detected using a SpectraMax i3x (Molecular Devices, San Jose, CA, USA) at 485 nm. The percentage of specific lysis was calculated with the equation [(sample–spontaneous)/(maximal–spontaneous) × 100]. Maximal lysis was obtained by incubating target cells with 1.5% Triton X-100, and spontaneous lysis was obtained by incubating target cells alone.

### 4.15. Serum Antibody and Cytokine ELISA

For detection of the MA-IAV-specific total IgG, IgG1, or IgG2a, mice serum samples were collected by centrifugation from heart-punctured blood. A total of 1 μg/mL of whole inactivated H1N1 MA-IAV was diluted in PBS and coated in each well of 96-well Nunc-Immunoplates (Thermo Fisher Scientific, Waltham, MA, USA) overnight at 4 °C. Serum samples were diluted in 2% BSA in PBS as a blocking buffer and treated in the coated plates for 2 h at room temperature. HRP-conjugated anti-mouse total IgG, IgG1, and IgG2a were diluted in the blocking buffer and incubated for an hour at room temperature.

For detection of collagen-specific IgG, on days 7, 45, and 52 after the initial type II collagen (CII, Sigma, St. Louis, MO, USA) vaccination, sera from inoculated mice were obtained. A total of 10 μg/mL of CII in PBS was coated onto Immunoplates overnight at 4 °C (Nunc). Serially diluted (1:5000–1:200,000) serum samples were placed in CII-coated wells and incubated for 2 h at room temperature after blocking with 5% bovine serum albumin in PBS, and then anti-mouse IgG-HRP (1:10,000) was added and incubated for 2 h at room temperature. Plates were developed with TMB after a series of PBST washes, and reactions were stopped by adding stop solution using an ELISA reader; the absorbance was determined at 450 nm (Bio-Rad, Hercules, CA, USA).

For the detection of OVA-specific IgG and IgM, mice sera were sampled at the indicated time points after immunization. A total of 100 μg/mL of OVA in PBS was coated onto Immunoplates overnight at 4 °C and ELISA analysis was performed similarly to that for CII.

Cultured supernatant cytokines, IFNγ and IL-4, were determined via an ELISA kit (BD Biosciences, San Jose, CA, USA). The enzyme reaction was performed with TMB substrate (BD Biosciences, San Jose, CA, USA) for 5 min, and the color development was stopped by 2N H_2_SO_4_. Optimal density was measured using a microplate reader model 680 (Bio-Rad, Hercules, CA, USA) at 450 nm.

### 4.16. IFNγ ELISPOT

The detection of IFNγ-expressing cells was determined via enzyme-linked immunospot assay (ELISPOT, BD Biosciences, California, San Jose). IFNγ-capture antibody was diluted in PBS (10 μg/mL) and coated in each well of an ELISPOT plate overnight at 4 °C. Then, each coated plate was blocked with RPMI 1640 supplemented with 10% FBS, 2 mM L-glutamine, 1% penicillin–streptomycin, 10 μg/mL gentamycin, and 50 μM β-mercaptoethanol (Gibco-BRL, Grand Island, NY, USA) for 2 h at room temperature. A total of 2 × 10^5^ re-stimulated splenocytes were prepared in RPMI 1640 culture medium and incubated in the coated plate for 18 h at 37 °C. Cells were then removed by washing with PBS and PBS containing 0.05% Tween-20 (PBST), and the biotinylated detection antibody diluted in 10% FBS in PBS (2 μg/mL) was added and incubated for 2 h at room temperature. After washing with PBST, HRP-conjugated streptavidin was added and incubated for an additional hour at room temperature. The enzyme reaction was carried out with AEC substrate (BD Biosciences, San Jose, CA, USA), and the reaction was stopped by washing with deionized water at the appropriate times, and the plate was dried for 2 h at room temperature. The number of spots was automatically measured using an ELISPOT plate reader (AID CLASSIC ELR08, AID, Bethesda, MD, USA).

### 4.17. Western Blot

Sorted cDCs were extracted in M-PER buffer (Thermo Fisher Scientific, Waltham, MA, USA) supplemented with a proteinase and phosphatase inhibitor cocktail (Sigma-Aldrich, St. Louis, MO, USA). Cell lysates were centrifuged at 13,000 rpm for 10 min at 4 °C, and the protein concentration in the supernatant was quantified by Bradford assay. Extracts were diluted in Laemmli sample buffer containing β-mercaptoethanol and boiled for 10 min. Proteins were resolved in 10% SDS–PAGE gel and then transferred to a PVDF membrane. Blotted membranes were incubated with primary antibodies against NF-κB p65, phospho-NF-κB p65 (Cell Signaling Technology, Danvers, MA, USA), and GAPDH (Santa Cruz Biotechnology, Santa Cruz, CA, USA), and then probed with 100 ng/mL of horse radish peroxidase (HRP)-conjugated anti-mouse (Sigma-Aldrich, St. Louis, MO, USA) or anti-rabbit-IgG secondary antibodies (Santa Cruz Biotechnology, Santa Cruz, CA, USA). Protein bands were visualized by using an LAS4000 mini (GE Healthcare, Chicago, IL, USA) and densitometric results were analyzed using the ImageJ program. The level of target protein was normalized to GAPDH.

### 4.18. CIA Induction and Clinical Scoring

Mice received a 100 µg intradermal (i.d.) injection of CII (Sigma-Aldrich, St. Louis, MO, USA) emulsified in 25 µL of complete Freund’s adjuvant (CFA) at the base of the tail. On day 14, mice received a second i.d. injection of 100 µg CII emulsified in incomplete Freund’s adjuvant (IFA). Mice were monitored daily for arthritis development.

Arthritis severity was graded on a 0–4 scale: 0 = normal paws, 1 = edema and erythema in one digit, 2 = slight edema or erythema in multiple digits, 3 = moderate edema involving the entire paw, and 4 = severe edema and erythema involving the entire paw. The average macroscopic score, with a maximum of 16, was calculated by summing the scores of all four paws.

### 4.19. Statistical Analysis

All statistical significance was determined using one- or two-way ANOVA testing with Bonferroni’s multiple comparison tests for between-group comparisons with Prism software (GraphPad Software 6.0, San Diego, CA, USA). The following symbols are used in the text, figures, and legends to indicate statistical significance: * *p* < 0.05, ** *p* < 0.01 *** *p* < 0.001, and **** *p* < 0.0001 and ns, not significant.

## Figures and Tables

**Figure 1 ijms-26-00149-f001:**
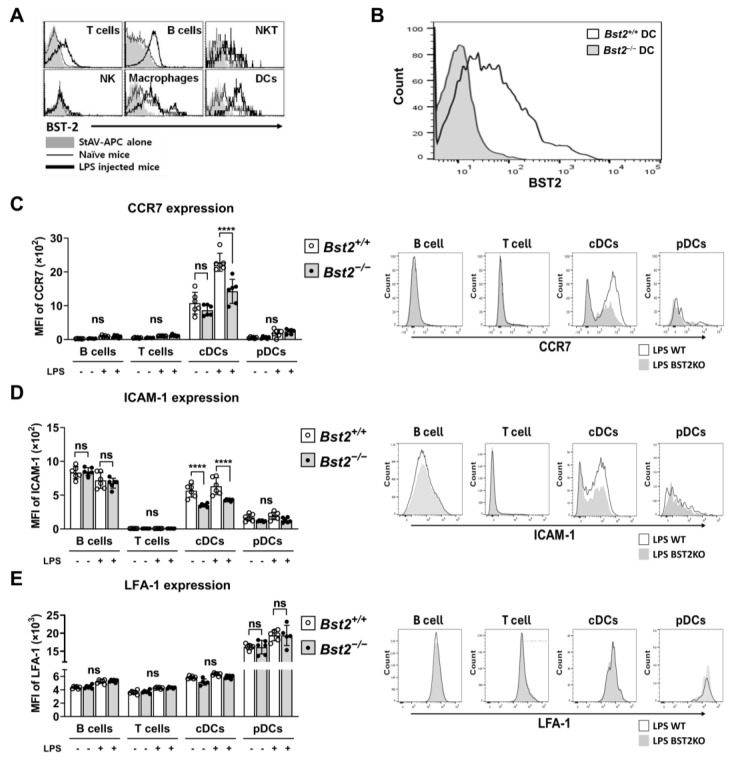
BST2 expression is involved in the expression of the chemokine receptor CCR7 and adhesion molecule ICAM-1 in cDCs. (**A**) BST2 expression in immune cells from the splenocytes of wild-type mice with or without LPS administration. (**B**) Purified cDCs from the splenocytes of *Bst2^+/+^* and *Bst2^−/−^* mice were stimulated for 24 h with 0.1 μg/mL of LPS and their BST2 expression was measured. (**C**–**E**) Splenocytes from *Bst2^+/+^* and *Bst2^−/−^* mice were stimulated as described in (**B**), and the expressions of CCR7 (**C**), ICAM-1 (**D**), and LFA-1 (**E**) in each gated population was analyzed by FACS. Immune cell subsets were gated using fluorochrome-conjugated cell surface marker-specific antibodies: B cells (CD3ε^−^CD19^+^), T cells (CD3ε^+^CD19^−^), cDCs (CD11c^high^CD11b^+^B220^−^), pDCs (CD11c^+^CD11b^−^B220^+^). Representative data are shown as mean ± S.D. from four independent experiments with similar results. **** *p* < 0.0001 and ns, not significant. The right panels in each staining show representative FACS images of LPS-stimulated samples.

**Figure 2 ijms-26-00149-f002:**
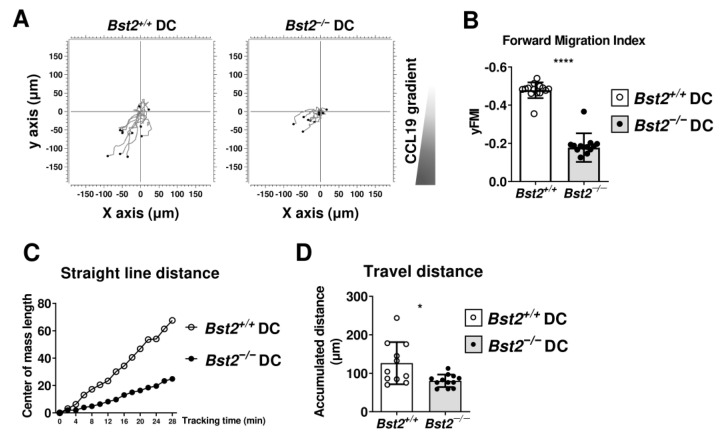
BST2 expression on cDCs is necessary for their migration through CCR7-mediated chemotaxis. (**A**) Trajectory plots were analyzed by tracking each of 15 Bst2^+/+^ and Bst2^−/−^ cDCs migrating toward a CCL19 gradient. Images were captured every 2 min for 1 h. (**B**) yFMI, (**C**) center of mass length, and (**D**) accumulated distance of Bst2^+/+^ and Bst2^−/−^ cDCs were measured in 15 slides. Representative data are shown as mean ± S.D. from at least three independent experiments with similar results. * *p* < 0.05 and **** *p* < 0.0001.

**Figure 3 ijms-26-00149-f003:**
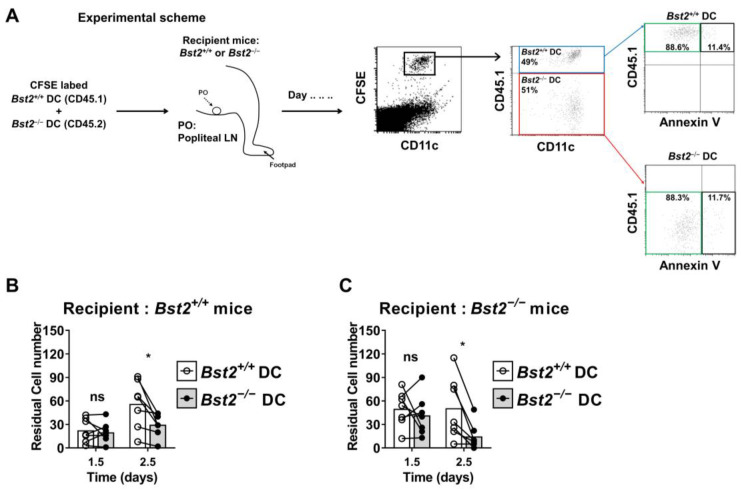
BST2 expression on cDCs facilitates their migration into draining lymph nodes. (**A**) Experimental scheme for in vivo DC migration. Equal numbers (5 × 10^5^ each) of CD45.1^+^*Bst2^+/+^* cDCs and CD45.2^+^*Bst2^−/−^* cDCs were mixed and stimulated with 1 μg/mL of LPS. After an hour of stimulation, cDCs were injected into the footpads of recipient mice and analyzed at 1.5 and 2.5 days post-transfer. The quality of cDCs was accessed before transfer by Annexin V staining. (**B**,**C**) Mixed cDCs were injected into *Bst2^+/+^* (**B**) or *Bst2^−/−^* (**C**) recipient mice, and the migrated DCs in dLN were analyzed by FACS and cell counting. The numbers on the y axis indicate the absolute number of cDCs in each dLN. Connected horizon bars indicate results from each individual recipient mouse. Representative data are shown as mean from at least three independent experiments with similar results. * *p* < 0.05 and ns, not significant.

**Figure 4 ijms-26-00149-f004:**
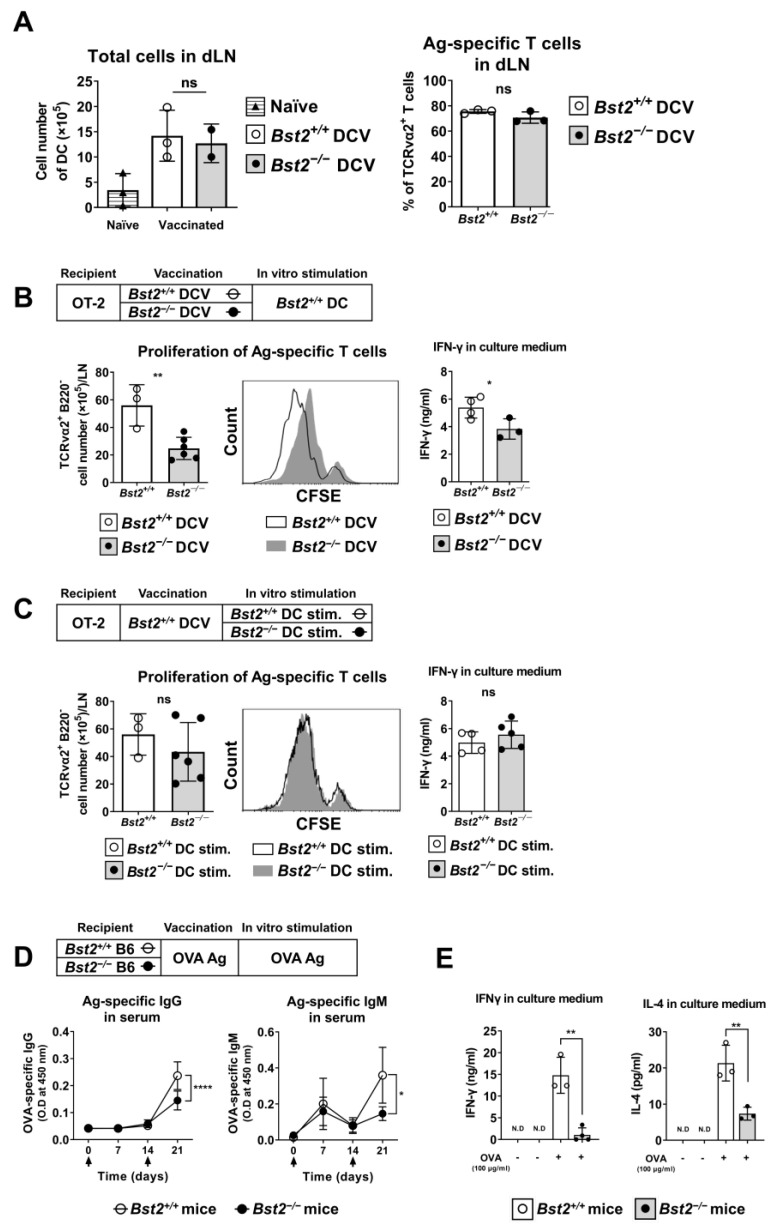
BST2-mediated migration is involved in overall immune responses. (**A**) OVA323-339-pulsed cDCs were injected into the footpads of OT-2 transgenic mice. The dLN cells were analyzed at 4 days post-transfer. Total cell number in the dLN (left) and the percentage of TCRVα2^+^ T cells (right) were analyzed by FACS and cell counting. (**B**) The dLN cells primed with Bst2^+/+^ and Bst2^−/−^ cDCs were labeled with CFSE and re-stimulated with OVA323-339-pulsed Bst2^+/+^ cDCs for 24 h. The amount of IFNγ in the cell culture supernatants was measured by ELISA. (**C**) The dLN cells primed with Bst2^+/+^ cDCs were labeled with CFSE and re-stimulated with OVA323-339-pulsed Bst2^+/+^ or Bst2^−/−^ cDCs for 24 h. The amount of IFNγ in the cultured supernatants was measured by ELISA. (**D**) After 3 weeks of immunization, OVA-specific IgG and IgM in the sera were measured by ELISA. The arrows on the horizontal axes indicate immunization points. (**E**) Whole splenocytes were re-stimulated with OVA protein for 24 h, and the amounts of IFNγ and IL-4 were measured by ELISA. Representative data are shown as mean ± S.D. from at least three independent experiments with similar results. * *p* < 0.05, ** *p* < 0.01, and **** *p* < 0.0001 and ns, not significant. ND, not detectable.

**Figure 5 ijms-26-00149-f005:**
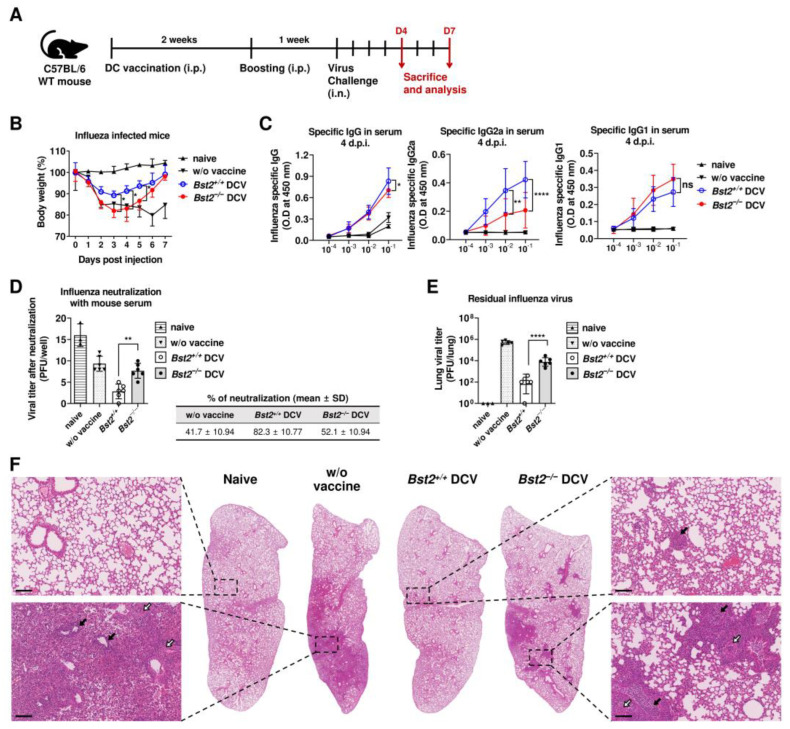
BST2-mediated DC migration is associated with optimal protection against MA-IAV. (**A**) Experimental scheme of DC vaccination. Bst2^+/+^ or Bst2^−/−^ cDCs sorted by MACS were incubated with 5 Multiplicity of Infection (MOI) of MA-IAV and 1 μg/mL of LPS and were injected intraperitoneally into Bst2^+/+^ C57BL/6 mice. (**B**) Body weights of MA-IAV-infected mice were measured daily for 7 days. (**C**) After 4 days of virus challenge, antibody titers of MA-IAV-specific total IgG, IgG2a, and IgG1 in the infected mice serum were measured by ELISA. (**D**) After 7 days of challenge, a neutralization assay against MA-IAV was performed using the mice serum. (**E**) After 4 days of challenge, virus titration of MA-IAV in the lungs was measured using a plaque assay. (**F**) Lung histology in vaccinated and infected mice. Representative micrographs of lung section stained with hematoxylin and eosin (H&E) are shown to determine global morphologic changes. Scale bar = 200 μm. Inflammation-based consolidation (open arrow) and cell infiltration (closed arrow) are indicated. Representative data are mean ± S.D. from at least three independent experiments (*n* = 3 recipients/naïve group, *n* = 6–7 recipients/infected group) with similar results. * *p* < 0.05, ** *p* < 0.01, and **** *p* < 0.0001 and ns, not significant.

**Figure 6 ijms-26-00149-f006:**
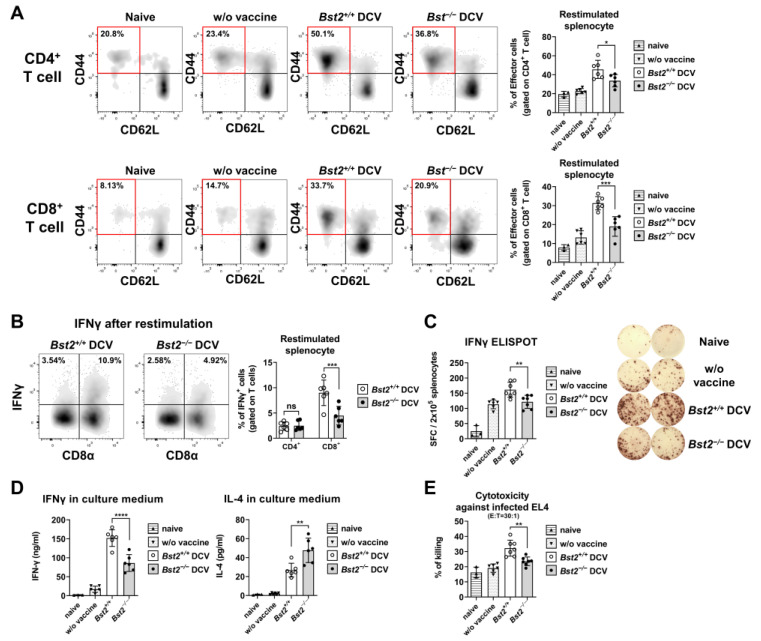
BST2-mediated DC migration is associated with the development of the effector T cell. Splenocytes from virus-infected mice were co-cultured with an equal number of MA-IAV-infected Bst2^+/+^ splenocytes. (**A**) After 3.5 days of re-stimulation, CD4+ (top) and CD8+ (bottom) T cells from the re-stimulated splenocytes were analyzed for effector T cell surface marker expression. The percentage of CD44^+^CD62L^−^ effector T cells was measured by FACS analysis. (**B**) After 1.5 days of re-stimulation, cultured splenocytes were treated with Golgistop, and the percentage of IFNγ-expressing T cells (CD3ε-gated) was determined by FACS analysis. (**C**) MA-IAV-specific cytokine secretion was measured by ELISPOT. (**D**) MA-IAV-specific cytokine secretion was also quantified by ELISA. (**E**) CTL response was measured using a calcein-AM release assay with MA-IAV-infected, calcein-labeled EL4 cells as targets. Re-stimulated splenocytes were co-cultured with target cells for 3 h. The percentage of specific lysis was calculated using the formula described in the Materials and Methods. Representative data are presented as mean ± S.D. from at least three independent experiments (**A**,**B**,**D**) or two independent experiments (**C**,**E**) with similar results; * *p* < 0.05, ** *p* < 0.01, and *** *p* < 0.001, **** *p* < 0.0001 and ns, not significant.

**Figure 7 ijms-26-00149-f007:**
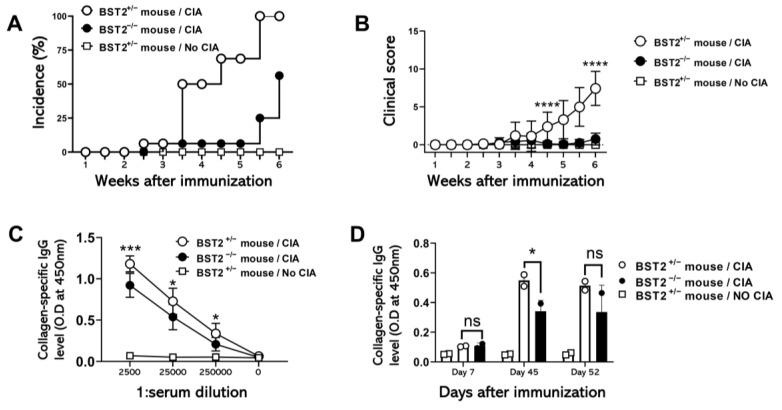
Clinical scores and collagen-specific IgG levels in the serum of each group of arthritic BST2-heterozygous and BST2-deficient mice. (**A**) Percentage of arthritic DBA1/J mice in each group. (**B**) Clinical scores of arthritis in DBA1/J BST2-heterozygous and BST2-deficient mice. Each paw was scored from 0 to 4 according to arthritis severity, with a maximum score of 16 per mouse. (**C**) ELISA evaluation of B cell response levels through collagen-specific IgG in each group of DBA1/J BST2-heterozygous and BST2-deficient mice. Samples were diluted from 1:2500 to 250,000 to evaluate the presence of collagen-reactive mouse IgG at day 45. (**D**) ELISA evaluation of collagen-reactive mouse IgG at days 7, 45, and 52. Samples were diluted 1:2500. * *p* < 0.05, *** *p* < 0.001, **** *p* < 0.0001 and ns, not significant.

## Data Availability

Data are contained within the article or Appendix A.

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
