# Peer review of "The Role of Bone Marrow Stromal Cell Antigen 2 (BST2) in the Migration of Dendritic Cells to Lymph Nodes"

_ijms, 2024, doi:10.3390/ijms26010149_

Round 1

Reviewer 1 Report

Comments and Suggestions for Authors

The manuscript of Sehoon Park et al. is a convincing summary about the work they had done to describe connections between tetherin expression of dendritic cells and their migration to lymph nodes, which have impact on antigen specific T cell activation and immune response.

Nevertheless, the manuscript has several smaller-larger imperfections. These involve smaller misspellings but also missing details needed to understand the article. I should also mention missing details that are important to the repeatability of the work.

I summarize these imperfections in mayor and minor concern lists:

Mayor concerns:

-          Representative figures  or description of flow cytometric gating strategies for the investigation of different leukocyte populations should be inluded in the Supplementary material.

-         Figure 1.A does not mention the source of the cells. They should be also splenocytes. The Fig.1.A subfigure legend is misleading. The StAV-APC control is correct, but the label of the untreated and LPS treated cells should be thin and thick lines to correspond the figure. The main legend of the figure mentions LPS administration, but it is not clear whether mice had LPS treatment or the cells in the cell cultures.

-          In most cases (both in the main text and in the Supplementary material), individual data points plotted on the bar charts indicate different independent experiment numbers than the figure legends mention. This reduce the credibility of the article. .Some figure legends resolve this problem with the sentence “at least … independent experiments”. You can mention in 4.19 Statistical analysis part, that the data points on the bar charts represent the independent experiment numbers, for clarity.

-          Figure 1.C-E: There is no description whether the cells of the right side FACS histograms are LPS-treated or non-treated cells.

-          Figure 2. There is no description about the represented results. What is “Forward migration index”? What is “Straight line distance”? What is “Travel distance”? At least the 4.4 Chemotaxis assay part should have some short description about it.

-          Figure 3. There is no description about the absolute or residual cell number determination in the manuscript.

-          line 184: The first mention of OT-2 transgenic mice. Some short description of these mice should be mentioned here or at least in the 4.1 part of the manuscript. A reference about them would be also useful.

-          line 188 and Figure 4. There are no details about what time after the priming the dLN cells were re-stimulated.

-          Figure 4.B-C. What were the durations of T cell restimulation at Figure 4.B and C? Do the bar charts of the left side represent cell numbers after restimulation? How were TCRvα2+ cell numbers evaluated?

-          line 203-204, Figure 4.E.: There is no details about this experiment in the Materials and Methods part.

-          line 203: “(Fig.4E)” should refer to Fig.4D

-          line 204: “(Fig.4F)” should refer to Fig.4E

-          There is also shift in the text in the case of Figure 6 references: lines 291-301: BàC, CàD, DàE, EàF. This is also present in the figure legend: lines 309, 311, 312

-          There is no indication whether the cells of Figure 6C are spleen cells, or gated on T cells. So the CD8α negative cells could be either mixed cells or CD4+ and double negative T cells.

-          line 457: “About 1.5~2.5 days later…” – in the text there are time points at day 1.5  AND 2.5!

-          line 463-464: “Sera or cells from immunized mice were analyzed three days after boosting immunization.” - Figure 4D suggest ELISA for day0, d6, d13, d21. No day14+3 indicated!

-          Please explain the meaning of MOI in the text at the first mention!

-          ELISA, ELISPOT, Western blot methods in section 4.15-4.17: Please indicate the concentration of the used reagents, or the dilution of the used reagent kit’s stock solutions.

Minor concerns in the main text:

-          line 18: “reduced” should also be put before “retention” for clarity.

-          line 49: “forms” instead of “formess”

-          line 66: “likeCCR7” – missing space

-          line 91: About “complete” loss of BTS2 expression on Fig.1B: The figure does not involve the control staining (e.g. FMO control) of the cells, so the completeness of the loss of expression can’t be seen.

-          line 118: “necessary “ is a too strong word for here. Bts2 -/- cells are able to migrate also, but to a lesser extent.

-          line 152: “enequal” à unequal

-          line 153: “apoptosid” à apoptosis

-          Figure 3: The usage of CD45 could be better instead of Ly5 (also in line 453)

-          line 219: “cultured supernatants” à cell culture supernatants

-          Figure 4.D: Do arrows on the horizontal axes indicates the first and the booster immunization times?

-          line 243: “Bst2-/- DC-vaccinated group experienced significantly more weight loss than did the 243 Bst2+/+ DC-vaccinated group” According to the figure, this is  true only in the first days after the infection. Instead, “Disease course was more severe in Bst2-/- mice.” could be a more suitable description here.

-          line 362: “retection” à retention

-          line 485: “LN2” à liquid nitrogen

-          Exponent numbers should be placed in superscript e.g.: line 469, line 488

-          line 524: “cytometry” à cytometers

-          line 530. line 560: IL-12 measurement was not mentioned in the manuscript

-          line 552,553: “CII” à type II collagen –please indicate the abbreviation and refer the distributor at first mention!

Minor concerns in the Supplementary material:

-          line 22: (Suppl.Figure 1. legend) CD11b is considered as a myeloid marker in mice. pDCs should not express it. “CD11b+” could be a mistyping. (Description/representation of gating strategies would help to avoid such problems.)

Author Response

Reviewer #1

The manuscript of Sehoon Park et al. is a convincing summary about the work they had done to describe connections between tetherin expression of dendritic cells and their migration to lymph nodes, which have impact on antigen specific T cell activation and immune response.

Nevertheless, the manuscript has several smaller-larger imperfections. These involve smaller misspellings but also missing details needed to understand the article. I should also mention missing details that are important to the repeatability of the work.

I summarize these imperfections in mayor and minor concern lists:

Mayor concerns:

Comment #1: Representative figures  or description of flow cytometric gating strategies for the investigation of different leukocyte populations should be inluded in the Supplementary material.

Answer: Thank you for the critical comment. We added below gating strategy for the cell populations in the supplementary Figure 1.   

Comment #2: Figure 1.A does not mention the source of the cells. They should be also splenocytes. The Fig.1.A subfigure legend is misleading. The StAV-APC control is correct, but the label of the untreated and LPS treated cells should be thin and thick lines to correspond the figure. The main legend of the figure mentions LPS administration, but it is not clear whether mice had LPS treatment or the cells in the cell cultures.

Answer: Thank you for the pointing out. Yes, the cells were splenocytes as the reviewer point out. We marked it in the figure legend. From “BST2 expression in immune cells from wild-type mice” to “BST2 expression in immune cells from splenocytes of wild-type mice”

We also corrected the sentence “ (B) BST2 expression in purified cDCs from the splenocytes of Bst2+/+ and Bst2-/- mice after 24 hour of stimulation with 0.1 μg/ml of LPS.” To “purified cDCs from the splenocytes of Bst2+/+ and Bst2-/- mice were stimulated for 24 hour with 0.1 μg/ml of LPS and their BST2 expression was measured.”

Comment #3: In most cases (both in the main text and in the Supplementary material), individual data points plotted on the bar charts indicate different independent experiment numbers than the figure legends mention. This reduce the credibility of the article. .Some figure legends resolve this problem with the sentence “at least … independent experiments”. You can mention in 4.19 Statistical analysis part, that the data points on the bar charts represent the independent experiment numbers, for clarity.

Answer: Thank you for your comment and sorry for any confusion. There may be some misunderstanding about multiple experiments. For the result in each figure, we've done several similar experiments, although not exactly the same each time. The results in the figures show the results of one representative experiment of them. Individual data points in the bar charts show individual values ​​from duplicate or triplicate wells obtained from multiple mice recruited to the single experiment. For example, Figure 1 shows a total of six readout obtained from three mice (splenocytes) in each genotype stimulated in two independent wells.         

Comment #4: Figure 1.C-E: There is no description whether the cells of the right side FACS histograms are LPS-treated or non-treated cells.

Answer: the shown histograms are from LPS treated group. We corrected the sentence “The right panels in each staining show representative FACS images” to “The right panels in each staining show representative FACS images of LPS stimulated sample”

Comment #5: Figure 2. There is no description about the represented results. What is “Forward migration index”? What is “Straight line distance”? What is “Travel distance”? At least the 4.4 Chemotaxis assay part should have some short description about it.

Answer: Thank you for the comment. we added the following sentence in the end of section 4.4. “The Forward Migration Index (FMI), commonly referred to as the chemotactic index, is a measure for directed, chemotactic cell migration. It represents the efficiency of the forward migration of cells. straight-line distance measures the distance of start and end point regardless of cells travel path. The center of mass represents the average of all single cell endpoints. Its x and y values indicate the direction, in which the group of cells primarily traveled.”

Comment #6: Figure 3. There is no description about the absolute or residual cell number determination in the manuscript.

Answer: we added injected DC numbers in the legend of figure 3.

Comment #7: line 184: The first mention of OT-2 transgenic mice. Some short description of these mice should be mentioned here or at least in the 4.1 part of the manuscript. A reference about them would be also useful.

Answer: Thanks for the comment. we added a short description for OT-2 in the text as follow. From “we transferred OVA323-339-pulsed Bst2+/+ or Bst2-/- cDCs into each footpad of OT-2 transgenic mice” to we transferred OVA323-339-pulsed Bst2+/+ or Bst2-/- cDCs into each footpad of ovalbumin-specific CD4+ T cell transgenic OT-2 mice.” We also added “Ova-specific CD4+ T cell transgenic OT-II mice were obtained from Jackson laboratory.” in the Materials and Methods section.

Comment #8: line 188 and Figure 4. There are no details about what time after the priming the dLN cells were re-stimulated.

Answer: Thank you and sorry for missing information. We corrected the legend of figure 4.B as followThe dLN cells primed with Bst2+/+ and Bst2-/- cDCs were labeled with CFSE and re-stimulated with OVA323-339-pulsed Bst2+/+ cDCs for 24 hours.”

Comment #9: Figure 4.B-C. What were the durations of T cell restimulation at Figure 4.B and C? Do the bar charts of the left side represent cell numbers after restimulation? How were TCRvα2+ cell numbers evaluated?

Answer: Restimulation duration in both Figures 4.B and 4.C were the same. We added restimulation time in each figure legends. Bar chart in the left side show post-stimulation results. TCRvα2+ cells were detected with anti-Vα2 and anti-Vβ5 antibodies. We missed these antibodies in the Materials and Methods section in the original text and added them in the revised manuscript.    

Comment #10: line 203-204, Figure 4.E.: There is no details about this experiment in the Materials and Methods part.

Answer: We added an assay detail for OVA-specific Abs in Materials and Methods section as follow.For detection of OVA-specific IgG and IgM, mice sera were sampled at the indicated time points after immunization. 100 μg/ml of OVA in PBS was coated onto Immunoplates overnight at 4 °C and ELISA analysis was performed similarly to CII.”

Comment #11: line 203: “(Fig.4E)” should refer to Fig.4D

Answer: Thank you for your thoughtful pointing out. We corrected it.

Comment #12: line 204: “(Fig.4F)” should refer to Fig.4E

Answer: Thank you for your thoughtful pointing out. We corrected it.

Comment #13: There is also shift in the text in the case of Figure 6 references: lines 291-301: BàC, CàD, DàE, EàF. This is also present in the figure legend: lines 309, 311, 312

Answer: Thank you for the comment. We should not mark Figure 6B in the original text. Both Fig6.A and 6.B altogether is Figure 6A in the revised Figure 6.

Comment #14: There is no indication whether the cells of Figure 6C are spleen cells, or gated on T cells. So the CD8α negative cells could be either mixed cells or CD4+ and double negative T cells.

Answer: Thanks for the critical comment. The population in Figure 6C (Figure 6B in revised manuscript) shows T cell gated through CD3e. We added an explanation in the figure legend as follow. From “the percentage of IFNγ-expressing T cells was determined by FACS analysis” to “the percentage of IFNγ-expressing T cells (CD3ε-gated) was determined by FACS analysis”

Comment #15: line 457: “About 1.5~2.5 days later…” – in the text there are time points at day 1.5  AND 2.5!

Answer: Thanks for the comment. We corrected the mistakes according to the reviewer’s comment.

Comment #16: line 463-464: “Sera or cells from immunized mice were analyzed three days after boosting immunization.” - Figure 4D suggest ELISA for day0, d6, d13, d21. No day14+3 indicated!

Answer: Thanks for pointing our mistake. We corrected the sentence "Sera or cells from immunized mice were analyzed three days after boosting immunization." to "Sera or cells from immunized mice were analyzed a week after boost immunization." Additionally, the dates in Figure 4D have been modified to 0, 7, 14, and 21 to accurately reflect the experimental timeline.

Comment #17: Please explain the meaning of MOI in the text at the first mention!

Answer: Thank you for your comment. The meaning of "Multiplicity of Infection (MOI)" has been added at the first mention in lines 278 and 484 for clarity.

Comment #18: ELISA, ELISPOT, Western blot methods in section 4.15-4.17: Please indicate the concentration of the used reagents, or the dilution of the used reagent kit’s stock solutions.

Answer: Thanks for the comment. we added the concentrations of reagents in the Materials and Methods Section.

Minor concerns in the main text:

#1: line 18: “reduced” should also be put before “retention” for clarity.

#2: line 49: “forms” instead of “formess”

#3: line 66: “likeCCR7” – missing space

#4: line 91: About “complete” loss of BTS2 expression on Fig.1B: The figure does not involve the control staining (e.g. FMO control) of the cells, so the completeness of the loss of expression can’t be seen.

Answer: Thanks for the pointing. We correct the mentioned sentence from “However, DCs from BST2-/- mice exhibited a complete loss of BST2 expression” to “However, DCs from BST2-/- mice exhibited no sign of BST2 upregulation”

#5: line 118: “necessary “ is a too strong word for here. Bts2 -/- cells are able to migrate also, but to a lesser extent.

Answer: Thanks for the comment. We agree the reviewer’s opinion. We replaced the title from “BST2 expression on cDCs is necessary for CCR7-mediated chemotaxis” to “BST2 expression on cDCs promotes CCR7-mediated chemotaxis”

#6: line 152: “enequal” à unequal

#7: line 153: “apoptosid” à apoptosis

#8: Figure 3: The usage of CD45 could be better instead of Ly5 (also in line 453)

Answer: Thanks for the comment. we replaced Ly5 to CD45.

#9: line 219: “cultured supernatants” à cell culture supernatants

#10: Figure 4.D: Do arrows on the horizontal axes indicates the first and the booster immunization times?

Answer: Yes, they are. We added following sentence in the figure legend of Fig. 4D. “The arrows on the horizontal axes indicate immunization points.”

#11:  line 243: “Bst2-/- DC-vaccinated group experienced significantly more weight loss than did the 243 Bst2+/+ DC-vaccinated group” According to the figure, this is  true only in the first days after the infection. Instead, “Disease course was more severe in Bst2-/- mice.” could be a more suitable description here.

Answer: Thank you for the comment. We agree to the reviewer’s opinion. We corrected the sentence in the text from “the Bst2-/- DC-vaccinated group experienced significantly more weight loss than did the Bst2+/+ DC-vaccinated group (Fig. 5B)” to “disease course was more severe in the Bst2-/- DC-vaccinated group than that of the Bst2+/+ DC-vaccinated group (Fig. 5B)”

#12: line 362: “retection” à retention

#13: line 485: “LN2” à liquid nitrogen

#14: Exponent numbers should be placed in superscript e.g.: line 469, line 488

#15: line 524: “cytometry” à cytometers

#16: line 530. line 560: IL-12 measurement was not mentioned in the manuscript

Answer: thanks for the pointing. We removed IL-12 in the text.

#17: line 552,553: “CII” à type II collagen –please indicate the abbreviation and refer the distributor at first mention!

Answer: Thank you and we added “type II collagen (CII, Sigma)”

  • All other minor comments the reviewer pointed above were corrected accordingly and marked red in the text.

Minor concerns in the Supplementary material:

#1:   line 22: (Suppl.Figure 1. legend) CD11b is considered as a myeloid marker in mice. pDCs should not express it. “CD11b+” could be a mistyping. (Description/representation of gating strategies would help to avoid such problems.)

Answer: Thanks for the valuable and critical comment. We added following gating strategy for Figure 1 and also corrected CD11b expression from CD11b+ to CD11b-

Reviewer 2 Report

Comments and Suggestions for Authors

Sehoon Park et al. demonstrated that BST2 regulates the expression of CCR7 and ICAM-1 in cDCs. Consequently, BST2 directs the effective migration and retention of cDCs. This study holds significant interest for me.

The overall findings of this study are novel and logically presented. Notably, the authors employ multiple in vivo models to validate the function of BST2 in cDCs regarding migration, which provides a robust and convincing basis for their conclusions. However, it remains unclear whether other cell types are involved, particularly considering BST2 is reported in the growth and development of B cells. Please refer to the detailed comments below for further clarification.

Major points:

1. Figure 1 clearly demonstrates that BST2 knockout limits LPS-induced CCR7 and ICAM-1 expression in cDCs but not in pDCs, B cells, or T cells. The conclusion is convincing, but LPS is not the typical stimulus used for T and B cell activation. Furthermore, it is intriguing to consider how CCR7 and ICAM-1 expression would be affected if pDCs are stimulated with TLR7/9 ligands.

2. The in vivo studies presented in Figures 3-7 provide evidence that BST2/CCR7-mediated cDC migration is associated with downstream immune responses, such as T cell development and IgG secretion. However, it is noteworthy that B cells could also be the target of BST2 in these in vivo models. The authors should clarify and discuss the role of B cells in these contexts.

3. Beyond cDC migration regulation, it is essential to investigate whether BST2 also participates in antigen presentation.

Minor points:

1. Is there any difference in BST2 function between cDC1 and cDC2 groups?

2. In Figure 5, I am curious about the IL-21 level in T cells. Is it possible that BST2+ DCs regulate B cells indirectly through the regulation of IL-21?

Author Response

Major points:

Comment #1: Figure 1 clearly demonstrates that BST2 knockout limits LPS-induced CCR7 and ICAM-1 expression in cDCs but not in pDCs, B cells, or T cells. The conclusion is convincing, but LPS is not the typical stimulus used for T and B cell activation. Furthermore, it is intriguing to consider how CCR7 and ICAM-1 expression would be affected if pDCs are stimulated with TLR7/9 ligands.

 Answer: Thank you for your thoughtful comment. We agree that LPS is not typically used as a stimulus for T and B cell activation. However, our aim in this study was to focus on the effects of BST2 knockout on cDCs, and LPS has been widely used to examine activation markers such as CCR7 and ICAM-1 in these cells. We acknowledge that pDCs, B cells, and T cells might respond differently to LPS, and the impact of BST2 knockout in these cell types is an interesting direction for future studies. As for your suggestion about pDCs being stimulated with TLR7/9 ligands, we also find this an intriguing avenue to explore. We will consider investigating the effects of BST2 knockout on CCR7 and ICAM-1 expression in pDCs under these conditions in future experiments.

Comment #2: The in vivo studies presented in Figures 3-7 provide evidence that BST2/CCR7-mediated cDC migration is associated with downstream immune responses, such as T cell development and IgG secretion. However, it is noteworthy that B cells could also be the target of BST2 in these in vivo models. The authors should clarify and discuss the role of B cells in these contexts.

 Answer: Thank you for your insightful comment. We agree that B cells could potentially be a target of BST2 in the in vivo models we used. While our study primarily focused on the role of BST2 in cDC migration and its impact on downstream immune responses, including T cell development and IgG secretion, we acknowledge that B cells may also play an important role in these processes. To clarify this possibility we revised the text as below.

“We then assessed the role of BST2 expression in humoral immunity. Bst2+/+ and Bst2-/- mice were immunized with OVA antigen, and OVA-specific IgG and IgM production were measured in the sera one week after boost immunization. We found substantially lower level of OVA-specific IgG and IgM antibodies in the sera of Bst2-/- mice compared to Bst2+/+ mice (Fig. 4D). Additionally, splenocytes from Bst2-/- mice produced less IFNγ and IL-4 upon OVA stimulation compared to splenocytes from Bst2+/+ mice (Fig. 4E).

These results suggest that BST2 expression on cDCs play a critical role in migration and retention of cDCs in the dLNs, thereby contributing to optimal immune responses including T cell priming and humoral immunity.” to

“We then assessed the role of BST2 expression in humoral immunity. Bst2+/+ and Bst2-/- mice were immunized with OVA antigen, and OVA-specific IgG and IgM production were measured in the sera one week after boost immunization. We found substantially lower level of OVA-specific IgG and IgM antibodies in the sera of Bst2-/- mice compared to Bst2+/+ mice (Fig. 4D). Insufficient antibody production in Bst2-/- mice compared to Bst2+/+ mice, may be due to a deficiency in T cell stimulation by Bst2-/- dDCs, as well as a defect in the Bst2-/- B cells themselves. Although we did not further anaylzed the role of Bst2 in B cells, Additional experiment showed that splenocytes from OVA-immunized Bst2-/- mice produced less IFNγ and IL-4 upon OVA stimulation compared to splenocytes from Bst2+/+ mice (Fig. 4E), which suggest at least some involvement of T cells.

These results suggest that BST2 expression on cDCs play a critical role in migration and retention of cDCs in the dLNs, thereby contributing to optimal immune responses including T cell priming and humoral immunity.”

Comment #3: Beyond cDC migration regulation, it is essential to investigate whether BST2 also participates in antigen presentation.

Answer: Thank you for your valuable comment. We agree that investigating the potential role of BST2 in antigen presentation is an important direction for future studies. While our current research primarily focuses on the regulation of cDC migration, we recognize the significance of exploring BST2's involvement in antigen presentation. We will consider addressing this aspect in future experiments to better understand the full scope of BST2's functions in immune responses. As far as we analyzed, however, no significant differences were found when dLN cells were stimulated with OVA323-339 pulsed Bst2+/+ or Bst2−/− cDCs (Figure 4C). These results demonstrate that the effect of BST2, at least on antigen presentation, is not strong.

Minor points:

#1: Is there any difference in BST2 function between cDC1 and cDC2 groups?

 Answer: Thank you for the comment. No analysis was performed regarding the different roles of cDC1 and cDC2. We will keep this in mind for further research.

#2: In Figure 5, I am curious about the IL-21 level in T cells. Is it possible that BST2+ DCs regulate B cells indirectly through the regulation of IL-21?

Answer: Thank you for the comment. We are sorry. We did not analyze IL-21 production during the study. We will keep this in mind in future studies.